# In Silico Screening of Metal-Organic Frameworks for Formaldehyde Capture with and without Humidity by Molecular Simulation

**DOI:** 10.3390/ijms232213672

**Published:** 2022-11-08

**Authors:** Wei Li, Tiangui Liang, Yuanchuang Lin, Weixiong Wu, Song Li

**Affiliations:** 1Energy and Electricity Research Center, Jinan University, Zhuhai 519070, China; 2School of Energy and Power Engineering, Huazhong University of Science and Technology, Wuhan 430074, China

**Keywords:** formaldehyde, metal-organic frameworks, high-throughput computational screening, grand canonical Monte Carlo simulation

## Abstract

Capturing formaldehydes (HCHO) from indoor air with porous adsorbents still faces challenges due to their low capacity and poor selectivity. Metal-organic frameworks (MOFs) with tunable pore properties were regarded as promising adsorbents for HCHO removal. However, the water presence in humid air heavily influences the formaldehyde capture performance due to the competition adsorption. To find suitable MOFs for formaldehyde capture and explore the relationship between MOFs structure and performance both in dry air and humid air, we performed grand canonical Monte Carlo (GCMC) molecular simulations to obtain working capacity and selectivity that evaluated the HCHO capture performance of MOFs without humidity. The results reveal that small pore size (~5 Å) and moderate heat of adsorption (40–50 kJ/mol) are favored for HCHO capture without water. It was found that the structure with a 3D cage instead of a 2D channel benefits the HCHO adsorption. Atoms in these high-performing MOFs should possess relatively small charges, and large Lennard-jones parameters were also preferred. Furthermore, it was indicated that Henry’s constant (K_H_) can reflect the HCHO adsorption performance without humidity, in which the optimal range is 10^−2^–10^1^. Hence, Henry’s constant selectivity of HCHO over water (SK_H_ HCHO/H_2_O) and HCHO over mixture components (H_2_O, N_2,_ and O_2_) was obtained to screen MOFs at an 80% humidity condition. It was suggested that SK_H_ for the mixture component overestimates the influence of N_2_ and O_2_, in which the top structures absorb a quantity of water in GCMC simulation, while SK_H_ HCHO/H_2_O can efficiently find high-performing MOFs for HCHO capture at humidity in low adsorption pressure. The ECATAT found in this work has 0.64 mol/kg working capacity, and barely adsorbs water during 0–1 bar, which is the promising candidate MOF for HCHO capture.

## 1. Introduction

Volatile organic compounds (VOCs) include a variety of chemicals, some of which may have short- and long-term adverse health effects. Among the most popular VOCs, formaldehyde (HCHO), is very allergenic and carcinogenic even at very low concentrations [1,2,3]. The World Health Organization (WHO) recommended that a safe concentration of formaldehyde vapor for humans must be below 0.08 ppm (30-min) and a threshold sensory irritation of 0.1 mg/m^3^, which also can be lethal at a concentration of 30 mg/m [4,5,6]. Therefore, the removal of formaldehyde from contaminated air or the industrial process is in demand. The various methods for abating HCHO from indoor areas include photocatalysis [7,8], catalytic oxidation [9,10], and adsorption-based [11,12], and formaldehyde capture has been one of the most promising strategies due to the facile operation [13]. To date, a number of adsorbents including activated carbons [14], zeolites [13], SiO_2_ [15,16], AlOOH [17], amine-supported materials [18], etc. have been explored for formaldehyde removal. However, these traditional adsorbents are not suitable for addressing the continuous release of formaldehyde due to their non-polarity and highly amorphous nature [6], especially for the low adsorption capacity and poor selectivity in humid air [14,19,20].

Metal-organic frameworks (MOFs), as an emerging class of porous crystalline materials composed of inorganic metal nodes and organic ligands, have attracted increasing research interest due to their high specific surface areas and pore volumes, well-defined porosity, and tunable pore properties, which have been utilized for gas storage, separation, and catalysis [21,22]. Many MOFs (ZIF-67 [23], ZIF-8 [24], UiO-66 [25]) exhibited strong affinity toward HCHO and were reported as chemical capacitance sensors to detect it with satisfactory responses at room temperature. The potential for removal of HCHO was also demonstrated from a number of case studies on diverse MOFs. Wang et al. [26] prepared α, β, γ-CD-MOFs to capture HCHO at room temperature, and found that γ-CD-MOFs can almost totally remove 0.5 mg/m^3^ HCHO within 15 min, which indicated the high efficiency of the MOFs adsorbent even in such a low concentration. Accordingly, diamine-appended MIL-101(Cr) with water resistibility was synthesized, and it was found that the highest adsorption capacity can reach 5.4 mol/kg in 150 ppm concentration. [27] Nevertheless, the strong competitive of water molecules in humid indoor air hinders the adsorption of HCHO by occupying the adsorption sites preferentially. It was noted that the formaldehyde capacity of UiO-66-NH_2_ from 27.67 mg/g without humidity decreased to 9.84 mg/g at 12% relative humidity [28]. It is essential to obtain the adsorption performance with and without humidity to investigate the competitive adsorption between water and the HCHO of MOFs.

The arrangement of metal nodes and organic linkers, as well as the self-assembly topology of structures, leads to a diversity of experimental MOFs [29]. Moreover, there are also millions of hypothetical MOFs (hMOFs) by computational design [30], which makes it extremely difficult to search for target MOFs applied in HCHO removal. High-throughput computational screening (HTCS) based on the grand canonical Monte Carlo (GCMC) has become an efficient strategy to quickly find suitable MOFs for adsorption and separation applications, including carbon capture [31], H_2_ storage [32], and H_2_S separation [22]. During the screening, the structure-property relationship can be extracted to guide the rational design of high-performance porous adsorbents. Our previous work [33] screened out Y-BTC, ZnCar, and Ni-BIC from 2932 kinds of Computational-ready, experimental (CoRE) MOFs, and indicated that it has better capture and regeneration performance compared with activated carbon in a high HCHO concentration. However, the correlation between the adsorption performance and MOF characteristics remains to be clarified, especially the inside information of atoms in high-performing MOFs. Regarding the competition adsorption between water and HCHO, Yuan et al. [12] recently identified hydrophilic and hydrophobic MOFs by the Henry’s constant (K_H_), referring to the value of ZIF-8. They regarded MOFs with K_H_ < 2.6 × 10^6^ mmol/(g·Pa) as hydrophobic and then evaluated the adsorption capacity and selectivity of 31,399 hMOFs without humidity by GCMC and machine learning. Hence, the competition adsorption behavior between water and HCHO, as well as the structure-property relationship for MOFs with humidity, await further investigation.

Considering the HCHO capture performance with and without humidity focus in this work, the capacity and selectivity of HCHO in N_2_ and O_2_ (dry air) were established and the structure-property relationship was extracted in dry air, including the charge and Lennard-jones parameter of atoms in MOFs. The K_H_ selectivity (SK_H_) was then used to screen high-performance MOFs in humid air, in which the adsorption isotherm at the ambient temperature was simulated to verify it. It was suggested that SK_H_ of HCHO over water can play a crucial role in screening suitable MOFs for HCHO capture in low pressure. This work offers the molecular understanding of the rational design of high-performing MOFs for HCHO removal both in dry and humid air.

## 2. Results and Discussion

### 2.1. HCHO Capture Performance without Humidity

In this work, GCMC simulation was carried out to evaluate the HCHO pressure swing adsorption (PSA) performance of 1668 CoRE-MOFs in dry air between 0.1 and 1 bar. Figure 1 shows the relationship of working capacity (ΔW), selectivity (S), and heat of adsorption (Q_st_) to the pore size. It is revealed that the highest working capacity is 4.01 mol/kg, the refcode in the Cambridge Structural Database (CSD) is LAVSUY, followed by the DUBWON (3.98 mol/kg), and PARMIG (3.93 mol/kg). It denotes LCD located in the range of 5–6 Å for most MOFs with ΔW > 2 mol/kg and selectivity over 10^3^, presented in Figure 1a. Notably, Bellat et al. [13]. previously reported the total uptake of 2.3 mol/kg (7 wt%) of Ga-MIL-53 at 2000 ppm, room temperature, and Wang et al. [27] found the adsorption uptake of 3.34 mol/kg in MIL-101(Cr) at 150 ppm and then can up to 5.49 mol/kg after being post-modified by ethylenediamine. The highest working capacity in this work is larger than the capacity of MOFs without modification in the experiment, which indicated that there are promising MOFs awaiting discovery. In addition, most MOFs with large working capacities (>2 mol/kg) have relatively small LCDs (4–6 Å), which is nearly double the formaldehyde dynamic diameter (2.43 Å) [34]. This may be ascribed to the suitable interaction of MOFs in such a pore size, which reflected in the moderate heat of desorption (40~50 kJ/mol). A similar phenomenon of pore size dependence for gas adsorption also can be found in Wilmer et al.’s [35]. CH_4_ storage work and Banerjee et al.’s [36] Xe/Kr separation work. Moreover, Figure 1b indicated that the selectivity fluctuates with the heat of desorption, while higher Q_st_ benefits from the increase of selectivity. When the LCD > 10 Å, it was found that most structures exhibit poor HCHO capture performance (ΔW < 2.0 mol/kg, S < 10^3^ and Q_st_ < 40 kJ/mol). However, as presented in Appendix A, it was noted that excessive strong interaction leads to the HCHO being difficult to desorb during the PSA process, which will be discussed later.

In indoor air, formaldehyde is generally found in trace amounts with very low partial pressure, which adsorbs in Henry’s law region. Thus, Figure 2a presented the correlation between adsorption performance and K_H_. It was found that the MOFs exhibited poor ΔW and S when K_H_ was small than 10^−2^ mol/(kg·Pa), which can be ascribed to the poor interaction that limited the HCHO adsorption. As for those structures with K_H_ > 10^1^ mol/(kg·Pa), it is suggested that strong interaction benefits the selectivity of HCHO over N_2_ and O_2_. However, as shown in Appendix A, the overlarge host-adsorbate interaction causes the HCHO extremely hard to desorb, which is not conducive to HCHO capture. It was indicated that 10^−2^ ≤ K_H_ ≤ 10^1^ was favored for high working capacity and moderate selectivity that benefits HCHO capture. Such a result suggests that K_H_ can be used to pre-screen adsorbents for HCHO capture in dry air. Moreover, the correlation between atom distribution in crystal (MaxER, MinER) and ΔW was also investigated. Notably, as shown in Appendix A, the MaxER = MinER = 0.33 indicated that the MOF crystal is cubic topology and x, y, z axisymmetric. For MinER = 0, the structures tend to be 2-D layers. Hence, when the MinER is close to 0, it means the pore of structures tends to be a channel instead of a cage. In Figure 2b, it was found that MaxER near 0.4 and MinER located in 0.2–0.3 are favored for high working capacity, while most structures with MaxER > 0.6 and MinER < 0.1 exhibit low working capacity. Compared with the channel, 3D cages with different properties in each direction were preferred in HCHO capture.

We further analyzed the correlation between adsorption performance and the force field of atoms in MOFs, including average positive/negative charge(APC/ANC) and LJ parameters. According to Figure 3a, APC < 0.2 (ANC > −0.2) and APC > 0.5 (ANC < −0.5) exhibit poor working capacity, which the APC located in 0.2–0.5 for most structures with ΔW > 0.2 mol/kg. As shown in Appendix A, it was suggested that the enhancement of LJ interaction always favors the increase in HCHO capacity. The excessive interaction makes the HCHO difficult to desorb from the MOFs in Appendix A, which led to a decrease in working capacity, similar to the tendency found in K_H_. Furthermore, as shown in Figure 3b, the selectivity of HCHO over N_2_ and O_2_ scatter in a wide range (1–10^7^), can be divided into three parts according to the LJ parameters. For those MOFs with 0 < Aε < 1.5 and 0 < Aσ < 0.12, they have large enough pore volume (Appendix A), but the weak interaction limited the adsorption of HCHO, which makes the selectivity lower than 10^4^. As for those structures with Aε > 3 and Aσ > 0.21, the tiny pore volume cannot afford higher capacity, which also makes it unsuitable for HCHO capture. Therefore, it was suggested that 1.5 ≤ Aε ≤ 3.0 and 0.12 ≤ Aσ ≤ 0.21 are beneficial to the enhancement of selectivity, and most structures with S > 10^5^ are located in this range. Moreover, it was also found that the high selectivity accompanied by satisfying capacity in Appendix A, which indicated the combined moderate charge and LJ parameters, favors the HCHO capture performance of MOFs.

The top 10 MOFs with excellent formaldehyde capture performance are listed in Table 1. Among them, the best MOF is LAVSUY, with 6.62 Å LCD, 0.43 MaxER, 0.50 e APC, 0.16 kcal/mol Aε, 1.18 × 10^−2^ mol/(kg·Pa) K_H_, which was predicted to have 4.01 mol/kg working capacity and 2722 selectivity. As shown in Appendix A, the LAVSUY has *bcu* (body-centered cubic) topology, Y nodes connected by 1,3,5-Benzenetricarboxylic acid. In addition, other top-performance MOFs exhibited similar structural characteristics. For example, LCD located in 4.25–6.62 Å, MaxER in 0.37–0.56, APC in 0.16–0.59, Aε in 0.13–0.23 kcal/mol, K_H_ in 7.37 × 10^−2^–2.68 × 10^−1^ mol/(kg·Pa), and other descriptors are provided in Appendix A, which is quite consistent with the suitable range for HCHO capture found in previous results.

The adsorption isotherm of HCHO, N_2,_ and O_2_ mixture components obtained from GCMC simulation for the top 3 MOFs (LAVSUY, DUBWON, and PARMIG) are presented in Figure 4a–c. All MOFs almost were Type I adsorption isotherm [37] defined by IUPAC and exhibited ultra-high capacity with extremely low N_2_ and O_2_ capacity. It is worthy of note that the DUBWON and PARMIG seem to reach the saturation capacity when the pressure is larger than 0.8 bar, whereas the LAVSUY probably tends to have a higher capacity as the pressure continues to increase. Moreover, combined with the snapshots of Appendix A, the density plots in Figure 4d–f illustrated that the HCHO majority adsorb in the center of the cage close to the metal nodes of MOFs, which is consistent with the results of Figure 2b.

### 2.2. HCHO Capture Performance with Humidity

As we mentioned before, the competitive adsorption between HCHO and H_2_O would heavily influence the HCHO capture performance in humid air. However, estimating the HCHO capture performance for a large quantity of MOFs via GCMC simulation or experiment is extremely time-consuming [38]. It was proposed that the K_H_ of water can be adapted to identify whether the MOFs are hydrophilic or hydrophobic in HCHO capture. Moreover, the results in dry air suggested that K_H_ are the dominant factor to determine the HCHO capture performance. Thus, regarding the heavy competition between water and HCHO, there are two Henry’s selectivity (SK_H_) were calculated to screen out suitable MOFs in humid air, type 1: HCHO over water, type 2: HCHO over water, N_2,_ and O_2_. As shown in Figure 5a, it was found that the LCD of the top 3 MOFs for SK_H_ HCHO/H_2_O is located in a wide range (5–13 Å). Whereas the small LCD (~5 Å) exhibited better performance for SK_H_ HCHO/(H_2_O + N_2_ + O_2_) in Figure 5b, similar to the trend found in dry air.

Figure 6 was presented to illustrate the relationship between Henry constant selectivity and chemical descriptor, including MPC, MNC, Aσ, and Aε. It was found that the SK_H_ HCHO/H_2_O depend significantly on the charge since they are nonpolar adsorbates. In Figure 6a, it was found that most SK_H_ HCHO/H_2_O > 10 MOFs with MPC < 2. As for MPC ≥ 2, a large quantity of MOFs exhibited SK_H_ HCHO/H_2_O < 10^−2^ due to the strong Coulombic interaction between MOFs and water. Moreover, as shown in Appendix A, it was suggested that top MOFs for SK_H_ HCHO/H_2_O exhibited low void fraction and high LJ descriptors (Aσ > 0.2 and Aε > 3), including ECAHAT (LCD~12 Å). Moreover, as shown in Figure 6b, high Aσ and Aε also benefit the increment of SK_H_ HCHO/(H_2_O + N_2_ + O_2_), which indicated that Lennard-jones interaction is a dominant role in determining the HCHO capture performance in humid air. Moreover, in Appendix A, it was found most MPC > 2 MOFs have extremely large K_H_ for water that is not favored both in SK_H_ HCHO/H_2_O and SK_H_ HCHO/(H_2_O + N_2_ + O_2_).

The LCD, MPC, MNC, and K_H_ of the top 10 MOFs for SK_H_ HCHO/H_2_O were provided in Table 2. JAVTAC has a maximum SK_H_ HCHO/H_2_O, which is 418.76, followed by WOJJOV (194.11), and ECAHAT (144.74). Notably, it was found that all the MOFs in Table 2 have extremely low K_H_ of N_2_ and O_2_, which indicated they probably have a poor affinity toward N_2_ and O_2_. Indeed, as shown in Appendix A, SK_H_ HCHO/(H_2_O + N_2_ + O_2_) is almost linear with the SK_H_ HCHO/H_2_O for those MOFs with K_H_ of water > 1. The ranking difference between SK_H_ HCHO/H_2_O and SK_H_ HCHO/(H_2_O + N_2_ + O_2_) majority are those MOFs with K_H_ of water < 1.

In order to verify Henry’s constant screening results, the GCMC simulations were implemented for six MOFs (JAVTAC, WOJJOV, ECAHAT, DORDUK, DOTTUC, and OHOMIH) to obtain the adsorption isotherm under 80% humidity conditions in 298 K. As shown in Figure 7d–f, all of the structures selected by SK_H_ HCHO/(H_2_O + N_2_ + O_2_) are highly hydrophilic structures that adsorb a lot of water (>4 mol/kg) in low pressure (0.1 bar). The JAVTAC has a higher formaldehyde uptake in lower pressure and is in agreement with Henry’s law. However, when the pressure gradually increases to 1.0 bar, the water molecules with polar functional groups occupy the adsorption sites preferentially, and the strong competitive adsorption of water molecules hinders the capture of HCHO [12]. The water uptake then exhibits an s-shaped isotherm and finally reaches a higher water loading in structures. As for WOJJOV, the water exhibited a similar trend with JAVTAC, while the HCHO uptake maintains at 0.9 mol/kg. For the ECAHAT, the water uptake is extremely low during the whole pressure range, and it has a 0.64 mol/kg working capacity between 0.1 bar and 1 bar, and 465 selectivity of HCHO over H_2_O, N_2,_ and O_2_, which is a promising candidate for HCHO capture under humidity conditions. In this study, it was indicated that SK_H_ HCHO/H_2_O can be recognized as a critical descriptor in low pressure. It remains a challenge to find a suitable descriptor for screening MOFs under humidity conditions in high pressure with reasonable computation cost.

## 3. Materials and Methods

### 3.1. MOFs Database

All MOF structures were obtained from the computation-ready, experimental (CoRE) MOF database Version 1.0 [39], which the solvent and disorder structures were removed from Cambridge structural database (CSD) by Chung and co-workers. The structure with density derived electrostatic and chemical (DDEC) [40] charges containing 2932 structures were developed by Nazarian [41]. After removing the structures with zero accessible surface area (ASA), there are 1668 structures to perform formaldehyde capture screening. The ASA, largest cavity diameter (LCD), pore limiting diameter (PLD), and available pore volume (V_a_) were computed using the 1.86 Å nitrogen probe in zeo++0.3. Helium void fraction (VF) and Henry’s constant of MOFs toward H_2_O, HCHO, N_2_ and O_2_ were obtained by the Widom particle insertion method.

### 3.2. Grand Canonical Monte Carlo

CoRE MOFs containing 1668 structures carrying DDEC charges were employed for high-throughput screening. GCMC simulations were implemented to obtain the adsorption performance of these structures in RASPA 2.0. During the screening stage, 4 × 10^4^ Monte Carlo cycles were performed to estimate the adsorption isotherms of each MOF, including the initial 2 × 10^4^ cycles of equilibration run, and the other 2 × 10^4^ cycles of the production run. Four Monte Carlo moves of insertion, deletion, rotation, and translation were implemented with equal probability. Identity change of adsorbate molecules for multi-component adsorption was performed with the two-fold probability of insertion, deletion, rotation, and translation moves. The simulation temperature was maintained at 298 K, and the pressure ranges from 0.1 bar to 1 bar with a molar ratio of HCHO:N_2_:O_2_ = 2:798:200. As for the GCMC simulation under the 80% humidity condition, a total of 4 × 10^6^ Monte Carlo cycles were carried out. The simulation of the molar ratio for HCHO:H_2_O:N_2_:O_2_ was set as 200/3280/77,216/19,304 to represent the 200 ppm HCHO concentration in the humid air. The adsorption performance, including working capacity (ΔW), selectivity (S) of HCHO in dry air, and K_H_ selectivity (SK_H_) of HCHO in humid air were computed using the following equation. The heat of adsorption of HCHO was obtained at 200 Pa in the pure component.
(1)$$\Delta W=W_{\mathrm{HCHO},1\mathrm{bar}}-W_{\mathrm{HCHO},0.1\mathrm{bar}}$$
(2)$$S=\frac{W_{\mathrm{HCHO},1\mathrm{bar}}/f_{\mathrm{HCHO}}}{\sum W_{i,1\mathrm{bar}}/f_{i}}$$
(3)$${\mathrm{SK}}_{H}=\frac{K_{H,\mathrm{HCHO}}/f_{\mathrm{HCHO}}}{\sum K_{H,i}/f_{i}}$$
where $W_{\mathrm{HCHO},1\mathrm{bar}}$ is the HCHO capacity at 298 K, 1 bar, $f_{i}$ is the fraction of gas component (HCHO, H_2_O, N_2_ and O_2_) in the mixture adsorbate.

### 3.3. Force Field

During molecular simulation, the Lennard-Jones (LJ) and Coulomb potentials were used to describe the non-bonded interactions between MOFs and adsorbates.
(4)$$V_{ij}=4\varepsilon_{ij}\left[{\left(\frac{\sigma_{ij}}{r_{ij}}\right)}^{12}-{\left(\frac{\sigma_{ij}}{r_{ij}}\right)}^{6}\right]+\frac{q_{i}q_{j}}{4\pi \varepsilon_{0}r_{ij}}$$

Herein, *ij* represents the two interacting atoms, where *ε* is the depth of the potential wall, $\sigma_{ij}$ is the finite distance at which the inter-particle potential is zero, $r_{ij}$ is the distance between the particles. All the LJ parameters of MOFs were taken from UFF force field [42], and the LJ parameters for N_2_ and O_2_ were adapted from the TraPPE force field [43]. The Lorentz-Berthelot mixing rule was applied for inter-atomic LJ interactions. $q_{i}$ and $q_{j}$ are the atomic partial charges of two interacting atoms, and $\varepsilon_{0}$ is the vacuum permittivity constant. Long-range Coulombic interaction was described by the Ewald method [44] with a cutoff of 12.8 Å. We calculate the average sigma of LJ interaction and the average epsilon of LJ interaction, represented as Aσ and Aε.

The water model we adapted is the Tip4p force field, for it can well represent the water adsorption property in hydrophobic MOFs [45]. The force field parameters of formaldehyde were taken from Hantal et al.’s study [46] in which the planar formaldehyde model was employed. The bond lengths of H-C and C=O are 1.101 and 1.203 Å, respectively, and the angle of H-C=O is 121.8°. In this model, only the C and O atoms carry fractional charges of +0.45 × 10^1^ and −0.45 × 10^1^, respectively, and a dipole moment of 2.6 D along C=O bond vector was applied. The N_2_ and O_2_ force field are taken from TraPPE. All of the parameters of adsorbates are summarized in Appendix A.

### 3.4. The Descriptor of MOF Characteristic

There are nine descriptors that were collected from the crystallographic information file (CIF) to describe the structural/energetic features of MOFs. LCD is defined as the diameter of the largest sphere that can fit in the pore of MOF. The MaxER and MinER is the maximum and minimum value of variance explained by principal component analysis (PCA) for atom distribution in three directions of the unit cell, in which MaxER = MinER = 0.33 stands for the isotropic crystal. MPC/MNC is the most positive/negative charge of atoms in a unit cell. As for APC/ANC, the average positive/negative charge per unit volume was calculated. Furthermore, Aσ and Aε is the average σ/ε of an atom in the Lennard-Jones interaction. These descriptors were verified to possess significant correlations with the HCHO capture performance of MOFs.

## 4. Conclusions

In this work, we perform high-throughput computational screening of CoRE MOFs for HCHO capture with and without humidity conditions. In the dry air, working capacity and selectivity were adopted to evaluate the HCHO capture performance. It was found that small pore size (5–6 Å) and moderate heat of adsorption (40–50 kJ/mol) are favored for HCHO capture. Such high-performing structures probably have a 3D cage instead of a 2D channel with moderate charge and Lennard-jones parameters (0.2 ≤ APC ≤ 0.5, 1.5 ≤ Aε ≤ 3.0, and 0.12 ≤ Aσ ≤ 0.21) that benefit to the HCHO adsorption. Moreover, it was indicated that K_H_ is the dominant factor to determine the HCHO capture performance, for which 10^−2^–10^1^ mol/(kg·Pa) is preferred. The density plot of HCHO adsorption and adsorption isotherm verified that the top3 working capacity MOFs (LAVSUY, DUBWON, and PARMIG) are suitable for the removal of HCHO without H_2_O’s existence.

The SK_H_ HCHO/H_2_O and SK_H_ HCHO/(H_2_O + N_2_ + O_2_) was then obtained to screen MOFs under humidity condition. It was found that SK_H_ HCHO/(H_2_O + N_2_ + O_2_) overestimates the influence of N_2_ and O_2_ ascribed to its high ratio in the air. It was suggested that MOFs with strong Coulombic interaction (high MPC, MNC) tends to have low SK_H_ HCHO/H_2_O whereas the large Lennard-jones parameters (Aσ > 0.2 and Aε > 3) are required for MOFs exhibiting high SK_H_ HCHO/H_2_O. Moreover, the adsorption isotherm of the top three structures (JAVTAC, WOJJOV, and ECAHAT) indicated that SK_H_ HCHO/H_2_O can be recognized as a critical descriptor in low pressure, which all structures barely adsorb water. The simulation suggested that ECAHAT was a promising candidate for HCHO capture under 80% humidity conditions in 1 bar, 298 K, which have 0.64 mol/kg working capacity and high selectivity (reach 465).

## Figures and Tables

**Figure 1 ijms-23-13672-f001:**
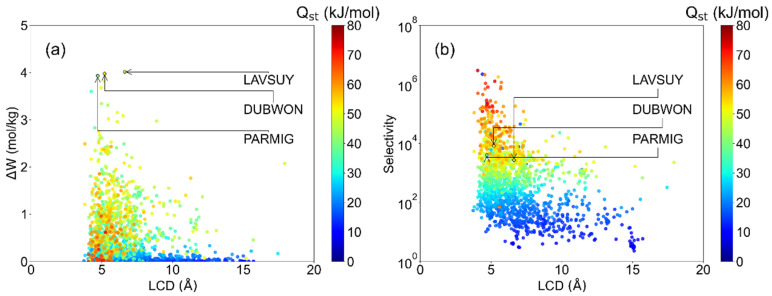
Relationship between the largest cavity diameter (LCD) and (**a**) HCHO working capacity, (**b**) selectivity, colored by the heat of adsorption (Q_st_).

**Figure 2 ijms-23-13672-f002:**
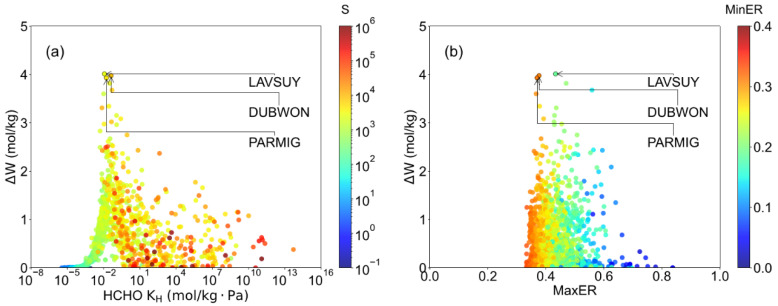
The relationship of formaldehyde capture performance and structural properties. (**a**) HCHO KH—working capacity relationship, colored by selectivity. (**b**) MaxER—working capacity relationship, colored by MinER.

**Figure 3 ijms-23-13672-f003:**
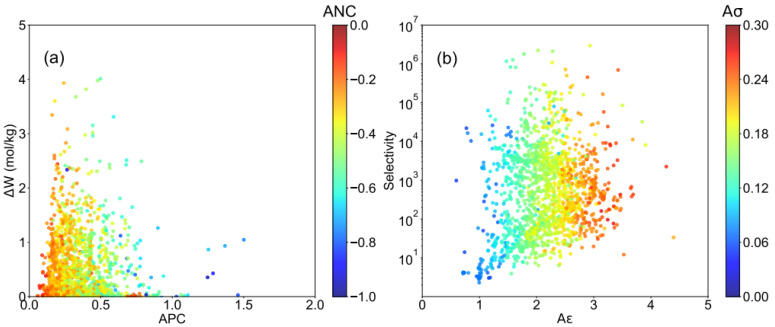
Relationship between descriptors and formaldehyde capture performance, (**a**) correlation between working capacity (ΔW), and average positive charge (APC) of MOFs, colored by average negative charge (ANC). (**b**) correlation between selectivity (S) and average ε (Aε) of MOFs, colored by average of σ (Aσ).

**Figure 4 ijms-23-13672-f004:**
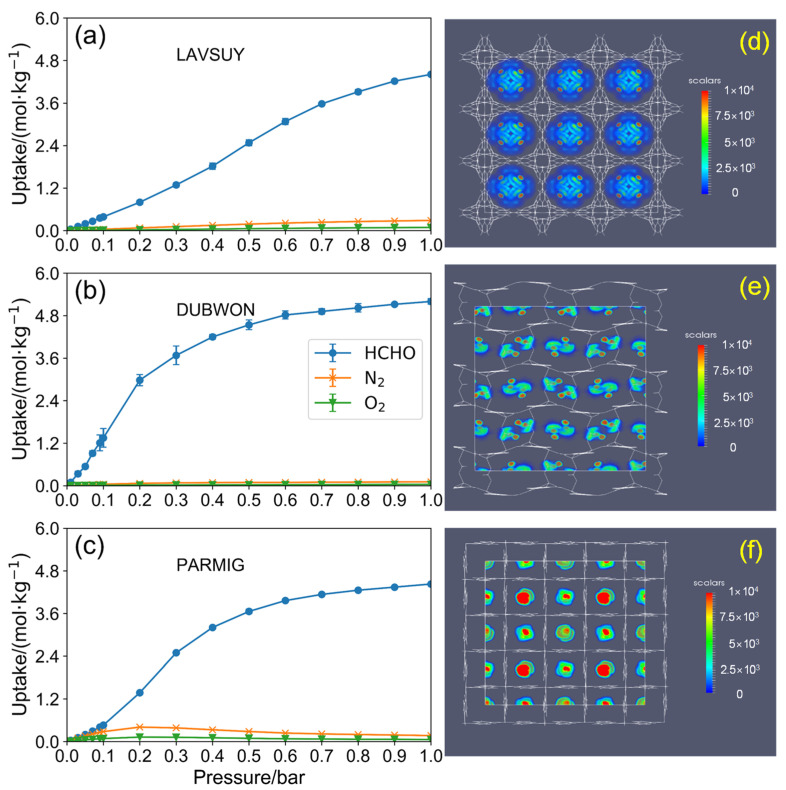
Formaldehyde, N_2_ and O_2_ adsorption isotherm of top-performing MOFs for a mixture component of HCHO/N_2_/O_2_ = 2/798/200 from GCMC simulation at 298 K ((**a**) for LAVSUY, (**b**) for DUBWON, (**c**) for PARMIG. The density distribution of formaldehyde adsorbates in (**d**) LAVSUY, (**e**) DUBWON and (**f**) PARMIG.

**Figure 5 ijms-23-13672-f005:**
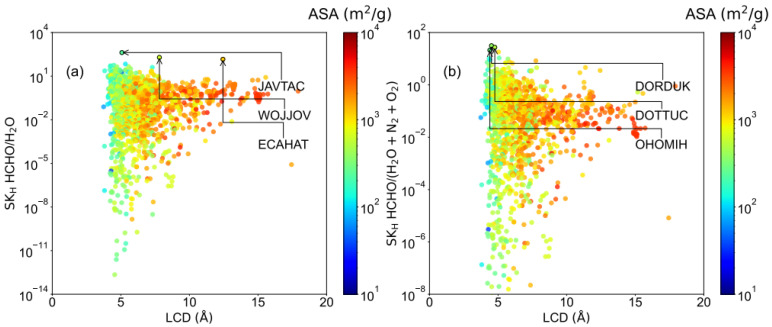
Relationship between LCD and (**a**) SK_H_ HCHO/H_2_O, (**b**) SK_H_ HCHO/(H_2_O + N_2_ + O_2_), colored by the ASA.

**Figure 6 ijms-23-13672-f006:**
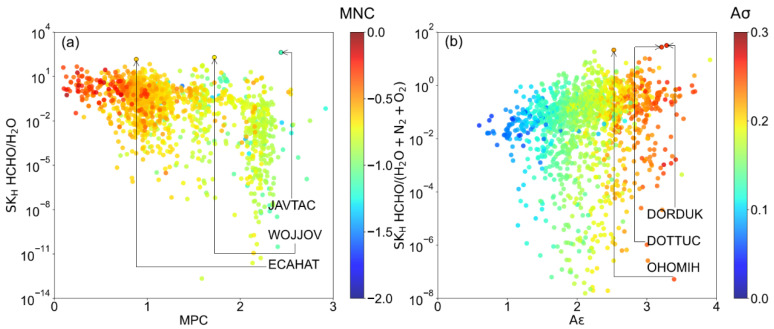
The relationship between chemical descriptors and Henry’s constant selectivity. (**a**) MPC and SK_H_ HCHO/H_2_O, colored by the MNC; (**b**) Aε and SK_H_ HCHO/(H_2_O + N_2_ + O_2_), colored by the Aσ.

**Figure 7 ijms-23-13672-f007:**
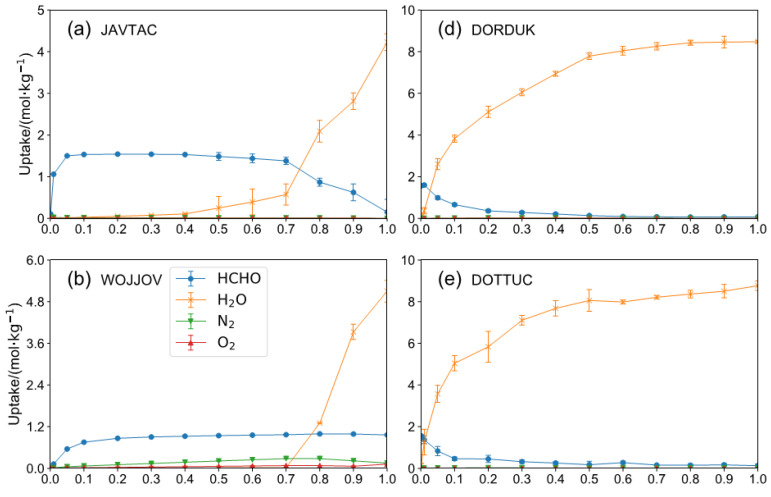
HCHO, H_2_O, N_2_, and O_2_ adsorption isotherm of top-performing henry constant selectivity MOFs (**a**) JAVTAC, (**b**) WOJJOV, (**c**) ECAHAT, (**d**) DORDUK, (**e**) DOTTUC, (**f**) OHOMIH for a mixture component of HCHO/H_2_O/N_2_/O_2_ = 200/3280/77,216/19,304 from GCMC simulation at 298 K.

**Table 1 ijms-23-13672-t001:** The largest cavity diameter (LCD), maximum explained ratio (MaxER), average positive charge (APC), average ε (Aε), Henry’s constant (KH), working capacity (ΔW) and selectivity (S) of the top 10 MOFs.

REFCODE	LCD Å	MaxER	APC e	Aε kcal/mol	KH mol/(kg·Pa)	ΔW mol/kg	S
LAVSUY	6.62	0.43	0.50	0.16	1.18 × 10^−2^	4.01	2722
DUBWON	5.20	0.38	0.48	0.14	4.19 × 10^−2^	3.98	8189
PARMIG	4.71	0.37	0.24	0.23	1.72 × 10^−2^	3.93	4044
SEHTAB	5.17	0.47	0.40	0.13	3.16 × 10^−2^	3.82	3157
DEYJIC	4.95	0.56	0.33	0.17	4.98 × 10^−2^	3.68	7689
ADIQEL	4.25	0.37	0.18	0.22	1.01 × 10^−2^	3.60	1453
LIFWOO	4.98	0.38	0.16	0.21	2.76 × 10^−2^	3.34	3486
DEFKUU	5.42	0.44	0.59	0.18	7.37 × 10^−3^	3.31	12,015
NABMUA01	6.10	0.43	0.44	0.18	1.17 × 10^−1^	3.15	2180
LOBHAM	6.51	0.39	0.27	0.17	2.68 × 10^−1^	3.09	4160

**Table 2 ijms-23-13672-t002:** The LCD, MPC, MNC, and K_H_ of HCHO, H_2_O, N_2_, and O_2_ for the top 10 MOFs in SK_H_ HCHO/H_2_O.

REFCODE	LCD Å	MPC e	MNC e	K_H_ HCHO mol/(kg·Pa)	K_H_ H_2_O mol/(kg·Pa)	K_H_ N_2_ mol/(kg·Pa)	K_H_ O_2_ mol/(kg·Pa)	SK_H_ HCHO/H_2_O
JAVTAC	5.08	2.44	−1.14	4.70 × 10^−1^	6.85 × 10^−5^	8.96 × 10^−5^	8.71 × 10^−5^	418.76
WOJJOV	7.81	1.72	−0.68	5.16 × 10^−2^	1.62 × 10^−5^	2.29 × 10^−5^	2.32 × 10^−5^	194.11
ECAHAT	12.44	0.88	−0.61	2.08 × 10^−2^	8.75 × 10^−6^	1.13 × 10^−5^	1.38 × 10^−5^	144.74
PUQYAC	5.33	1.03	−0.62	2.10 × 10^−3^	1.88 × 10^−6^	8.87 × 10^−6^	9.35 × 10^−6^	68.04
LIDZUV	4.49	1.6	−0.78	1.02 × 10^−2^	1.10 × 10^−5^	2.10 × 10^−5^	2.23 × 10^−5^	56.81
ZERQOE	4.24	1.59	−0.78	4.56 × 10^−3^	5.39 × 10^−6^	1.50 × 10^−5^	1.93 × 10^−5^	51.59
KAXQOR	4.23	1.59	−0.78	4.07 × 10^−3^	5.22 × 10^−6^	1.41 × 10^−5^	1.83 × 10^−5^	47.56
IXISOX	5.57	0.24	−0.36	3.84 × 10^−2^	5.12 × 10^−5^	9.73 × 10^−6^	1.04 × 10^−5^	45.73
PARMIG	4.71	0.76	−0.58	1.72 × 10^−2^	2.35 × 10^−5^	4.37 × 10^−5^	5.31 × 10^−5^	44.6
GUPBEZ	7.29	0.1	−0.31	3.72 × 10^−3^	5.64 × 10^−6^	2.01 × 10^−6^	2.26 × 10^−6^	40.24

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
