# Peer review of "In Silico Screening of Metal-Organic Frameworks for Formaldehyde Capture with and without Humidity by Molecular Simulation"

_ijms, 2022, doi:10.3390/ijms232213672_

Round 1

Reviewer 1 Report

The authors apply the Grand Ensemble Monte Carlo approach to a database of metal-organic frameworks (MOFs) using force field models for the MOFs and and guest species to explore the affinity and selectivity for formaldehyde capture.

They identify candidate MOFs with high working capacity for formaldehyde capture and relate the working capacity of thousands of MOFs to various descriptors of their structures. They also screen MOFs for high formaldehyde selectivity in high-humidity atmospheres, identify candidate MOFs and further explore the selectivity vs. pressure for the most promising systems.

This paper is a substantial work that provides a good example of the combination of databases and computational materials science to screen materials for applications, and actually identifies materials that may be useful for direct formaldehyde removal for air.

There are only a few minor suggestions for improvement:

(1) Line 14-15 "In order to find.."  : Sentence fragment. Some words are missing.

(2) Line 93: humidity air -> humid air

(3) Line 167: LCD  : would help to define the acronym again

(4) Lines 194; 198, etc.: "was suggested", "suggested". : If these are previous results, add citations. If they are current results, use present tense "Is suggested", "suggest"...

(5) Line 312: "lots of water".  Vague. Could you add a quantitative amount for what you mean by "lots of water"?

Author Response

Thank you very much or your positive comments! We have carefully revised the manuscript and provided the point-to-point responses according to your comments below. We hope our revised manuscript is acceptable for publication.

Reviewer 2 Report

The authors in this work calculates the HCHO adsorption from a series of MOF materials that are present in the Core database. They also performed the ability of these structures to separate HCHO from a mixture. All these structures were done under dry and humid conditions. The type of the simulations they used were the appropriate for such kind of simulations. The parameters for the interactions were taken from UFF force field whereas the charges were included in the cif files. The analysis that was done to justify the results is satisfactory.

To my knowledge there are not similar studies dealing with the adsorption and/or separation HCHO.

I think this study is carefully done and the discussion seems to explain the results. I suggest this study to be published without any revisions.

Author Response

Thank you very much for your positive comments! 
